# Kallikrein-11, in Association with Coiled-Coil Domain Containing 25, as a Potential Prognostic Marker for Cholangiocarcinoma with Lymph Node Metastasis

**DOI:** 10.3390/molecules26113105

**Published:** 2021-05-22

**Authors:** Saeranee Siriphak, Ravinnipa Chanakankun, Tanakorn Proungvitaya, Sittiruk Roytrakul, Doungdean Tummanatsakun, Wunchana Seubwai, Molin Wongwattanakul, Siriporn Proungvitaya

**Affiliations:** 1Centre of Research and Development of Medical Diagnostic Laboratories, Faculty of Associated Medical Sciences, KhonKaen University, Khon Kaen 40002, Thailand; ssaeranee@kkumail.com (S.S.); c.ravinnipa@gmail.com (R.C.); tanakorn@kku.ac.th (T.P.); pui_ddlab41@hotmail.com (D.T.); moliwo@kku.ac.th (M.W.); 2National Center for Genetic Engineering and Biotechnology, National Science and Technology Development Agency, Pathum Thani 12120, Thailand; sittiruk@biotec.or.th; 3Cholangiocarcinoma Research Institute, Faculty of Medicine, Khon Kaen University, Khon Kaen 40002, Thailand; wunchanas@yahoo.com; 4Department of Forensic Medicine, Faculty of Medicine, Khon Kaen University, Khon Kaen 40002, Thailand

**Keywords:** cholangiocarcinoma, CCDC25, KLK11, immunohistochemistry, prognostic marker

## Abstract

Cholangiocarcinoma (CCA) is a malignancy arising from cholangiocytes. Currently, the treatment and prognosis for CCA are mostly poor. Recently, we have reported that coiled-coil domain containing 25 (CCDC25) protein level in the sera may be a diagnostic marker for CCA. Subsequently, we identified three binding proteins of CCDC25 and found that kallikrein-11 (KLK11) expression was highest among those binding proteins. In this study, we investigated CCDC25 and KLK11 expression in CCA and adjacent normal tissues (*n* = 18) using immunohistochemistry. The results demonstrated that the expressions of CCDC25 and KLK11 in CCA tissues were both significantly higher than the adjacent tissues (*p* < 0.001 and *p* = 0.001, respectively). Then, using GEPIA bioinformatics analysis, KLK11 mRNA was significantly overexpressed in CCA tumor tissues compared with normal tissues (*p* < 0.05). Moreover, CCDC25 expression was positively correlated with KLK11 expression in CCA with lymph node metastasis (*p* = 0.028, r = 0.593). An analysis for the interaction of KLK11 with CCDC25 and other proteins, using STRING version 11.0, revealed that CCDC25 and KLK11 correlated with metastasis-related proteins. In addition, Kaplan-Meier survival curve analysis revealed that a high expression of KLK11 was associated with the poor prognosis of CCA. In conclusion, KLK11 is, as a binding protein for CCDC25, possibly involved in the metastatic process of CCA. KLK11 may be used as a prognostic marker for CCA.

## 1. Introduction

Cholangiocarcinoma (CCA) is a malignancy arising from cholangiocytes (bile duct epithelial cells). The disease is highly endemic in Southeast Asia, especially in the Khon Kaen Province, Northeast Thailand [1]. Many factors, such as benign biliary tract disease, primary sclerosing cholangitis, etc., are assumed to be involved in CCA genesis. Liver fluke, *Opisthorchis viverrini* (OV), infection is a common cause of CCA in Northeast Thailand because of the consumption of undercooked fish, the intermediate host of OV, and poor hygiene. Liver fluke infection in association with some carcinogens such as nitrosamine can cause chronic inflammation of cholangiocytes, leading to oxidative DNA damage and subsequent CCA transformation [2]. Nowadays the treatment for CCA is limited and the prognosis of CCA is mostly poor. Carbohydrate antigen 19-9 (CA 19-9) and carcinoembryonic antigen (CEA) have been used as serum biomarkers for routine screening for CCA, but their sensitivity and specificity are not satisfactory. By now, specific tumor markers for CCA are still lacking [3,4] and biopsy is the only way to reach a definite diagnosis of CCA [5]. Surgery is the best curative treatment; however, management is challenging because many patients are asymptomatic at the early stage and most of the patients are diagnosed at the advanced stage. Thus, the majority of CCA cases are unresectable and the recurrence after resection is common [5].

Recently, we found that CCDC25 is overexpressed in CCA tissues [6] and serum CCDC25 level of CCA patients is significantly higher than that of healthy controls [7]. Moreover, recently one of our colleagues (R.C.) identified three binding proteins of CCDC25, including kallikrein-11 (KLK11), family with sequence similarity 105, member A (FAM105A), and phosphatidylinositol 4-phosphate 5-kinase type-1 gamma (PIP5K1C). Among those CCDC25 binding protein candidates, KLK11 protein expression was the highest in CCA tissue lysates [8]. Therefore, in this study, the expression and correlation of CCDC25 and KLK11 in CCA tissues in association with clinical features were examined.

## 2. Results

### 2.1. Expression of CCDC25 and KLK11 in Cholangiocarcinoma Tissue

Among three CCDC25 binding protein candidates, KLK11 was the highest in protein expression intensity in CCA tissues (Appendix A) [8]. Thus, CCDC25 and KLK11 expression in CCA tissues were examined using immunohistochemistry. For both the CCDC25 and KLK11 stains, the results showed that the normal cholangiocytes monolayer of the intrahepatic bile ducts in the adjacent areas were stained with a light-brown color (Figure 1C,E), whereas the CCA cells were stained with a dark-brown color (Figure 1D,F). Staining of CCDC25 and KLK11 were seen in the cytoplasm of both cancerous and normal cells. As shown in Figure 2, the scatter plot revealed that the median H-score of CCDC25 and KLK11 in cancerous tissues were significantly higher than those of the adjacent tissues (*p* < 0.001 and *p* = 0.001, respectively).

### 2.2. Expression of CCDC25 and KLK11 in Cholangiocarcinoma with or without Lymph Node Metastasis

Since the expression of either KLK11 or CCDC25 was associated with the metastasis of colorectal cancers [9,10,11,12,13], the expressions of CCDC25 and KLK11 in CCA with or without lymph node metastasis were analyzed. The results show that the expressions of both CCDC25 and KLK11 were significantly higher in cancerous tissues than in the adjacent tissues of CCA with lymph node metastasis (*p* < 0.001 and *p* = 0.018, respectively) (Figure 3A,C), but not in the CCA without lymph node metastasis (*p* = 0.057 and *p* = 0.057, respectively) (Figure 3B,D).

### 2.3. Correlation between CCDC25 and KLK11 Expression in Cancerous Tissues

The correlation between the CCDC25 and KLK11 expressions in the cancerous tissues of CCA with/without lymph node metastasis was analyzed using Spearman’s test. The results of overall patients showed no correlation between KLK11 and CCDC25 expressions (*p* = 0.294, r = 0.236) (Figure 4A). However, a significant correlation was observed between KLK11 and CCDC25 expressions in CCA with lymph node metastasis (*p* = 0.028 and r = 0.593) (Figure 4B), but not in CCA without lymph node metastasis (*p* = 0.417 and r = 0.600) (Figure 4C).

### 2.4. Associations between CCDC25 and KLK11 Expression in CCA Tissues with the Overall Survival Time of Patients

To assess whether CCDC25 and KLK11 can be the potential prognostic markers, the association between CCDC25 and KLK11 expression with the survival time of CCA patients was examined. The CCA patients were divided into low- and high-expression groups of CCDC25 and KLK11, using median values as the cut-off of CCDC25 and KLK11 (155 and 185, respectively). The association between CCDC25 and KLK11 and overall survival time were determined using the Kaplan–Meier analysis. The results demonstrated that the overall survival time of CCA patients with a high expression of KLK11 was significantly shorter than those with a low expression of KLK11 (414 vs. 809 days, respectively; *p* = 0.048) (Figure 5B). By contrast, the overall survival time of the CCA patients was not significantly different (519 vs. 409 days, respectively; *p* = 0.566) between high and low expression of CCDC25 (Figure 5A).

### 2.5. CCDC25 and KLK11 mRNA Expressions in CCA Tissue and Overall Survival Time of CCA Patients Using GEPIA

GEPIA bioinformatics analysis was used to assess the correlation between the mRNA expression levels of CCDC25 and KLK11 in CCA tissues and the overall survival time of the CCA patients. The results show that KLK11 mRNA expression was significantly higher in cancerous tissue than in normal tissue (*p* < 0.05) (Figure 6B) but no difference was observed in CCDC25 mRNA expression (Figure 6A). Survival analysis revealed that both CCDC25 and KLK11 were not significantly statistically different with a 95% confidence interval. However, the overall survival of KLK11 was significantly longer in the low-expression group with a 90% confidence interval (*p* = 0.056) (Figure 7B), but not CCDC25 (Figure 7A).

### 2.6. Potential Protein–Protein Interaction Analysis

To speculate the potential roles of KLK11 and CCDC25 in CCA metastasis, the interaction of CCDC25, KLK11, and metastasis-related proteins were assessed using STRING version 11. The results showed that CCDC25 indirectly interacted with KLK11, and both CCDC25 and KLK11 were associated with metastasis-related proteins such as Janus kinase 2 (JAK2), epidermal growth factor receptor (EGFR), mammalian target of rapamycin (mTOR), transforming growth factor beta 1 (TGFβ1), vascular endothelial growth factor (VEGF), mothers against decapentaplegic 2 (SMAD2), and matrix metallopeptidase 9 (MMP9) (Figure 8).

## 3. Discussion

CCDC25 is a membrane-bound protein and acted as a target molecule for the neutrophil extracellular traps (NETs)-mediated metastasis of cancer cells [10,11,12]. NET DNA was reported as being associated with cancer metastasis [14]. The deletion or knockout of CCDC25 in breast cancer cells and colon cancer cells led to the inhibition of NET-mediated lung and liver metastases [10,11,12]. The overexpression of CCDC25 enhanced the metastasis of breast cancer cells [10,11]. Moreover, a high expression of CCDC25 in primary breast and colon cancer cells was associated with a poor prognosis of the patients [10,11]. Thus, in this study, CCDC25 and CCDC25 binding protein candidate expressions in CCA were investigated. KLK11 was chosen as the protein of interest because one of the researchers of our group showed that the expression of the KLK11 protein was the highest among the three other CCDC25 binding candidate proteins identified.

In this study, immunohistochemical staining demonstrated that the expressions of CCDC25 and KLK11 were significantly higher in cancerous than adjacent tissues of the CCA specimen. Furthermore, their expressions were higher in lymph node metastatic but not in non-lymph node metastatic CCA. Moreover, CCDC25 expression was correlated with KLK11 expression in lymph node metastatic CCA, as suggested by Yang et al. (2020), in that the silencing of CCDC25 in human breast and colon cancer cells can inhibit metastasis to the lungs and liver [11]. In this study, GEPIA software was used to predict the mRNA expression levels of KLK11 in CCA. The results show that KLK11 expression was significantly higher in cancerous than in normal tissue (*p* < 0.05). KLK11 is a trypsin-like serine protease encoded by the KLK11 gene located on chromosome 19, 19q13.41 [15,16]. KLK11 is involved in progression of various cancers. KLK11 expression was associated with lymph node metastasis in low rectal carcinoma [13]. The inhibition of KLK11 significantly suppressed tumor growth and induced apoptosis of colorectal cancer cells [15]. Likewise, this study revealed that the expression of CCDC25 together with the expression of KLK11 were correlated with lymph node metastasis in CCA patients.

In the present study, by Kaplan–Meier analysis, the estimated overall survival of CCA patients with higher KLK11 expression was significantly shorter than that of patients with lower KLK11 expression. In agreement with our results, higher levels of KLK11 expression in gastric carcinoma were associated with poor overall survival [16]. Also, the patients with low KLK11 expression showed better survival rates than those with high expression in low rectal carcinoma [13]. The patients with high KLK11 expression were associated with shorter survival in metastatic colorectal cancer [9] and ovarian cancer [17].

Although KLK11 was identified as the CCDC25 binding protein candidate, the STRING interaction analysis revealed that CCDC25 indirectly interacted with KLK11 via complement component 1, q subcomponent binding protein (C1QBP), and kininogen 1 (KNG1). This possible indirect interaction may account for the discrepancy between the correlation of KLK11 and CCDC25 to the lifespan of the CCA patients, in that a high expression of KLK11 was associated with a shortened life span, but a higher CCDC25 expression was not. Since C1QBP and KNG1 are supposed to be involved in the crosstalk between KLK11 and CCDC25, their role in tumor metastasis should be explored. There is a report that the silencing of C1QBP inhibited hepatic metastasis of pancreatic cancer cells in vivo [18]. The overexpression of C1QBP was associated with distant metastasis of breast cancer [19]. KNG1 was significantly overexpressed in colorectal cancer cells compared to normal and dysplastic tissues [20]. Furthermore, CCDC25 and KLK11 are predicted to interact with other transcription factors including JAK2, EGFR, mTOR, TGFβ1, VEGF, SMAD2, and MMP9. These proteins were reported as multifunctional proteins in cancers including metastasis. Actually, EGFR and VEGF are known as receptors that transduce multiple downstream signaling pathways in several cancers [21]. EGFR, as a receptor, acts as signal transduction for the PI3K/Akt and MAPK pathways in lung cancer [22]. In the EGFR-related signaling pathway, PI3K/Akt, MAPK, and mTor were involved in metastatic colorectal cancer [23]. EGFR is a receptor tyrosine kinase upstream of the PI3K–Akt–mTOR pathway in head and neck squamous cell carcinomas [24]. These pathways are involved in cellular processes including cell growth, survival regulation, and metabolism in multiple solid tumors. VEGF was activated by HIF1 and IGF2 in the tumor angiogenesis process, which led to tumor progression and metastases [21]. VEGF was released in glioblastoma by TGFβ regulation, leading to the phosphorylation of SMAD2, SMAD3, and SMAD1/5/8 and increased VEGF release [25]. TGFβ acts as a mediator in liver tumor progression and invasion [26]. TROP2 activates the JAK2/STAT3 pathway to promote metastasis in glioblastoma cells [27]. Furthermore, KLK11 plays a role in tumor progression and metastasis via the insulin-like growth factor (IGF) signaling pathway [28]. MMP9 was identified to mediate circ0001361-induced cell migration and invasion in bladder cancer [29], and miR-5692a induced tumor progression in hepatocellular carcinoma [30]. The expression of MMP9 was associated with lymph node metastasis in CCA [31]. The silencing of MMP9 reduced the effects of TP73-AS1 overexpression on cell invasion and migration in ovarian cancer [32]. In addition, in this study, STRING analysis revealed that CCDC25 was associated with integrin-linked kinase (ILK), as was suggested that the deletion of the intracellular C terminus of CCDC25 inhibited its interaction with ILK [11]. The silencing of ILK in breast cancer cells significantly decreased liver metastases [11]. The knockout of CCDC25 and ILK or β-parvin reduced the activation of RAC1 and CDC42 [11]. These data suggest that the ILK–β-parvin–RAC1–CDC42 cascade acts downstream of CCDC25 to mediate NET DNA stimulated liver metastasis [10,11]. Furthermore, after TGFβ1 treatment, Rictor expression is increased, and a complex is formed between Rictor and ILK. ILK and Rictor interact to regulate TGFβ1-induced EMT [33]. Thus, these molecules might interact with CCDC25 and KLK11 in association with the metastatic process. Further study is necessary to substantiate the possible roles of those molecules and signal transduction pathways in CCA metastasis.

## 4. Materials and Methods

### 4.1. Patients and Specimens

Paraffin-embedded CCA tumor tissue specimens from 18 patients were provided by Cholangiocarcinoma Research Institute (CARI), Faculty of Medicine, Khon Kaen University, Thailand. The patients underwent surgical operation at the Srinagarind Hospital, Khon Kaen University, between 2010 and 2012. According to the reports from pathologists, 14 of 18 patients were positive while 4 patients were negative for lymph node metastasis. This study was approved by the Human Ethics Committee of Khon Kaen University (approval no. HE631336) and written informed consent was obtained from each of the participants. Sample size of CCA tissue was calculated by G*Power software (v.3.1.9.2, the G*Power team, Universität Düsseldorf, Düsseldorf, Germany) [34] and using the median and quartile deviation of preliminary study by using immunohistochemistry assay.

### 4.2. Immunohistochemistry

To validate CCDC25 and KLK11 protein expression in CCA, 18 paraffin-embedded CCA sections were deparaffined by soaking in xylene for 5 min 3 times and then twice each of absolute, 95% and 70% ethanol for 2 min. The sections were boiled with 1X citrate buffer (pH 6.0) for 10 min and washed in 1X PBS-T buffer. Then, the sections were incubated with 3% H_2_O_2_ in methanol for 1 h in the dark to block endogenous peroxidase activity followed by incubation with 20% fetal bovine serum for 2 h to block nonspecific background binding. After that, the sections were incubated with 150 μL of anti-CCDC25 and anti-KLK11 antibody (Cat#orb2517 and orb373440, respectively, Biorbyt, Cambridge, UK) with a dilution of 1:400 and 1:150, respectively at 4 °C overnight. The sections were washed in 1X PBS-T and then incubated with 150 μL of anti-rabbit IgG peroxidase antibody for 1 h. Peroxidase activity was detected by diamino-benzidine for 5 min in the dark. Then, the sections were washed with running water and counterstained with hematoxylin for 5 min and washed with running water again. The sections were dehydrated by soaking in the ascending series ethanol and cleared in xylene. Finally, the sections were sealed with a cover glass using Permount [35].

### 4.3. Immunohistochemical Evaluation of CCDC25 and KLK11 Expressions in CCA Tissues

IHC results of CCDC25 and KLK11 were assessed microscopically at 400× magnification and the H-score was averaged from 10 examination fields. For the semi-quantitative assessment of IHC, the intensity of staining (0 = no staining; 1+ = weak staining; 2+ = moderate staining; and 3+ = strong staining) and percentage of tumor cells staining (0–100%) were graded. H-score was calculated as follows: H-score = (% of positively stained tumor cells at weak intensity × 1) + (% of positively stained tumor cells at moderate intensity × 2) + (% of positively stained tumor cells at strong intensity × 3). Thus, the range of H-score varies from 0 to 300 for each field. Finally, the H-score of 10 fields was averaged [36].

### 4.4. Evaluation of CCDC25 and KLK11 Expression in Tissue and Overall Survival Time

The mRNA expression levels of CCDC25 and KLK11 in CCA tissues and overall survival time were analyzed using gene expression profiling interactive analysis (GEPIA; http://gepia.cancer-pku.cn/, Beijing, China, accessed on 5 November 2020) [37]. In brief, the gene name was input into a text box and then clicked at “Boxplots”. The page showed datasets selection (cancer name), and then added cancer type and clicked to plot. The page showed the boxplot of RNA expression between cancer and normal tissues. For overall survival analysis, gene name was input into a text box of gene name and then clicked at “overall survival”. The output page results were represented as Kaplan–Meier curves.

### 4.5. Prediction of CCDC25 and Signaling Pathway

To predict the CCDC25- and KLK11-related signaling pathways, the potential interaction between CCDC25, KLK11, and metastasis-related proteins were analyzed using search tool for retrieval of interacting genes/proteins (STRING; v. 11.0; http://string.embl.db.org/, ELIXIR, UK, accessed on 20 January 2021) [38] In brief, CCDC25, KLK11, and metastasis-related proteins were input into a multiple name box. Then, “*Homo sapiens*” was selected as the organism, and then we clicked to continue. The page showed the name list of proteins and we clicked to continue. The page showed the confidence view. Stronger associations were presented as thicker lines. Weak associations were represented by thin lines and protein–protein interactions were presented as solid lines.

### 4.6. Statistical Analysis

The data were presented as the median. The difference between two non-parametric groups was estimated using the Mann–Whitney U test. The Mann-Whitney U test was used to compare the CCDC25 and KLK11 H-score in cancerous and adjacent normal tissues. The correlation between CCDC25 H-score and KLK11 H-score was analyzed using Spearman’s test. Kaplan–Meier analysis was used to estimate the overall survival time. GraphPad Prism software (version 8; GraphPad Software Inc., La Jolla, CA, USA) and SPSS software (version 20; SPSS, Inc., IBM, NY, USA) were used for statistical analyses, and *p* < 0.05 and *p* < 0.1 were considered to indicate a statistically significant difference.

## 5. Conclusions

In conclusion, we found that CCDC25 together with KLK11 expression were correlated with the lymph node metastasis of CCA. Furthermore, a higher KLK11 expression was associated with a poor overall survival time of CCA patients, it was firstly reported that the expression of KLK11 in CCA tissue can be a prognostic marker of CCA. The potential signaling pathway of CCDC25 was hypothesized to be associated with KLK11 involved in the metastatic process. Therefore, the certain role of CCDC25 and other CCDC25 partner proteins should be investigated in a further study.

## Figures and Tables

**Figure 1 molecules-26-03105-f001:**
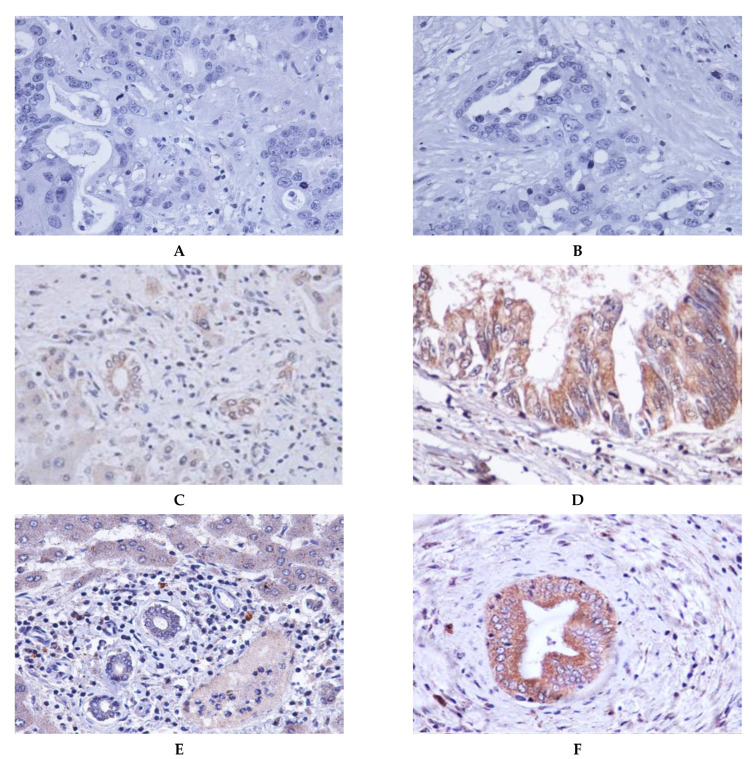
Representative immunohistochemical staining of CCDC25 and KLK11 in human CCA tissues (magnification, ×400). (**A**,**B**) Negative controls; (**C**) stained CCDC25 in adjacent non-cancerous tissue of a normal bile duct; (**D**) CCDC25 in CCA tumor cells; (**E**) stained KLK11 in adjacent non-cancerous tissue of a normal bile duct; (**F**) stained KLK11 in CCA tumor cells.

**Figure 2 molecules-26-03105-f002:**
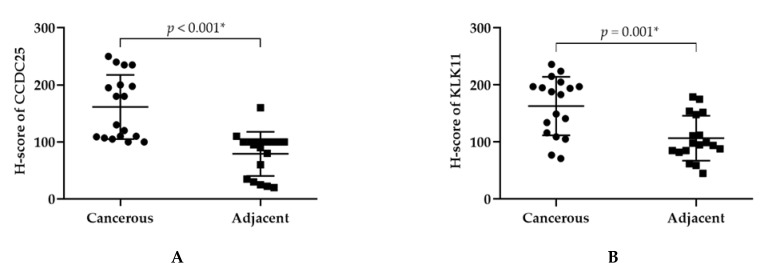
H-scores of CCDC25 and KLK11 in CCA tissues by using immunohistochemistry. (**A**) H-score of CCDC25. (**B**) H-score of KLK11. Mann–Whitney U test was used to compare H-scores between cancerous and adjacent tissues of CCDC25 and KLK11. * Statistical significance (*p* < 0.05).

**Figure 3 molecules-26-03105-f003:**
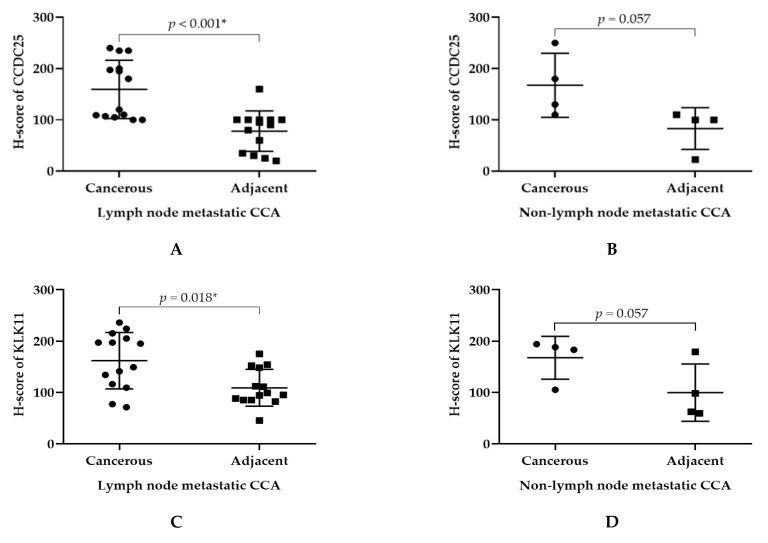
H-score of CCDC25 and KLK11 in lymph node metastatic CCA and non-metastatic CCA. (**A**) H-score of CCDC25 in lymph node metastatic CCA. (**B**) H-score of CCDC25 in non-lymph node metastatic CCA. (**C**) H-score of KLK11 in lymph node metastatic CCA. (**D**) H-score of KLK11 in non-lymph node metastatic CCA. Mann–Whitney U test was used to compare H-score between cancerous and adjacent tissues of CCDC25 and KLK11. * Statistical significance (*p* < 0.05).

**Figure 4 molecules-26-03105-f004:**
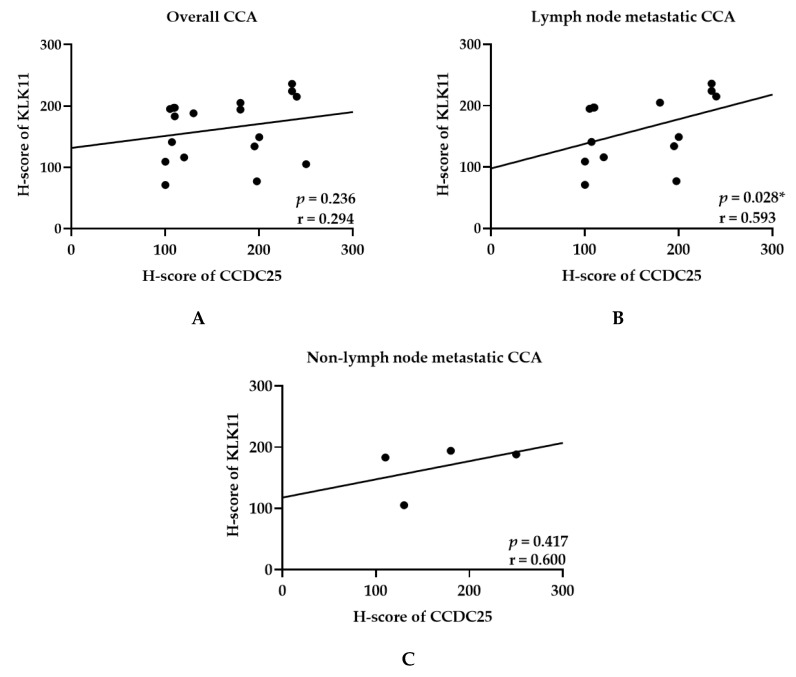
Correlation between CCDC25 and KLK11 H-score in cancerous tissues of (**A**) overall CCA patients, (**B**) CCA patients with lymph node metastasis, (**C**) CCA patients without lymph node metastasis. * Statistical significance (*p* < 0.05).

**Figure 5 molecules-26-03105-f005:**
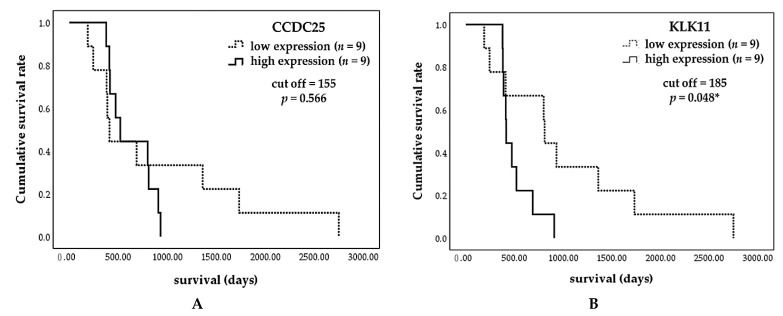
Kaplan–Meier survival curves of CCA patients based on H-score of CCDC25 and KLK11. CCA patients were divided into low- and high-expression using median values as cut-off for CCDC25 and KLK11 (155 and 185, respectively). The curves show overall survival of CCA patients having high expression (solid line) and low expression (dashed line). The survival times were observed between low- and high-expression groups of (**A**) CCDC25 and (**B**) KLK11 (log-rank test; *p* = 0.566 and *p* = 0.048, respectively). * Statistical significance (*p* < 0.05).

**Figure 6 molecules-26-03105-f006:**
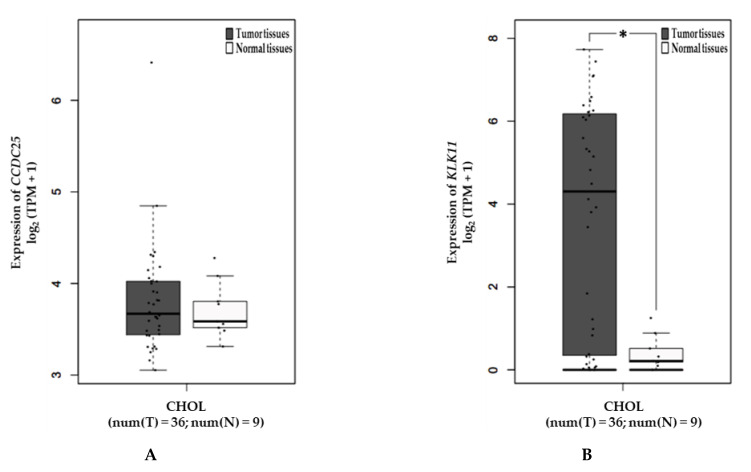
Expression of CCDC25 and KLK11 mRNA in CCA tissues analyzed by using GEPIA. (**A**) Expression of CCDC25 in CCA. (**B**) Expression of KLK11 in CCA. * Statistical significance (*p* < 0.05).

**Figure 7 molecules-26-03105-f007:**
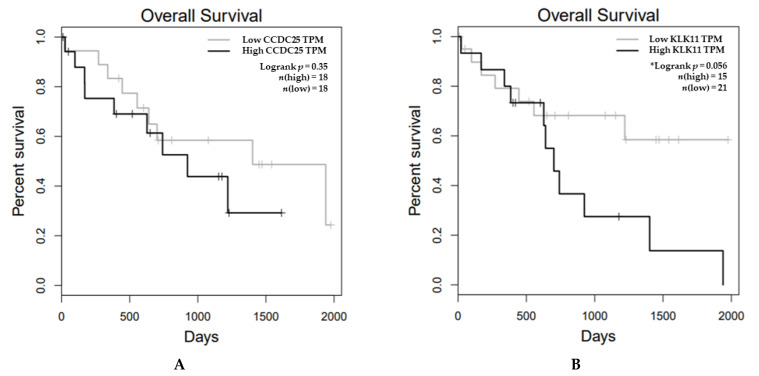
Kaplan–Meier survival curve analysis of CCDC25 and KLK11 mRNA expression levels and overall survival time of the samples obtained from GEPIA. The curves show overall survival of CCA patients having high expression (black line) and low expression (grey line). The survival time of CCA patients between low- and high-expression groups of (**A**) CCDC25 and (**B**) KLK11. * Statistical significance (*p* < 0.1).

**Figure 8 molecules-26-03105-f008:**
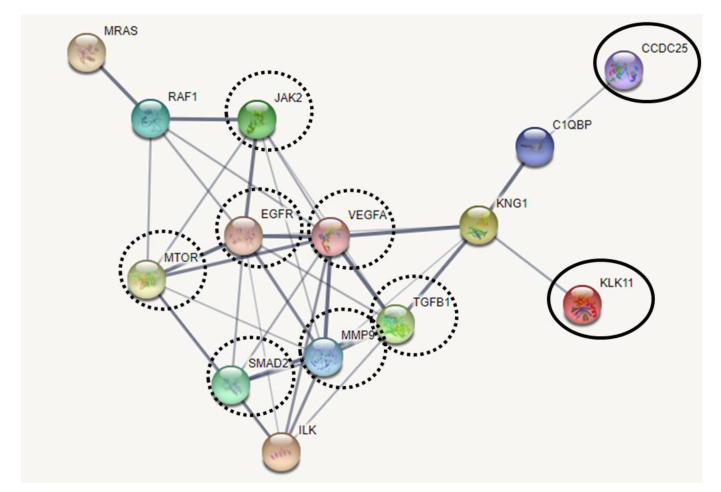
The interaction map of CCDC25, KLK11, and metastasis-related proteins. Protein–ligand interaction was predicted by STRING version 11.0. Protein–protein interactions were represented by solid lines. Stronger associations were represented by thicker lines. Weak associations were represented by thin lines. CCDC25 and KLK11 were represented in solid circles. Metastasis-related proteins were represented by dashed circles.

## Data Availability

Not applicable.

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
