# Peer review of "Kallikrein-11, in Association with Coiled-Coil Domain Containing 25, as a Potential Prognostic Marker for Cholangiocarcinoma with Lymph Node Metastasis"

_molecules, 2021, doi:10.3390/molecules26113105_

Round 1

Reviewer 1 Report

In this manuscript the authors described that CCDC25 together with KLK11 expression were correlated with lymph node metastasis of CCA and that higher KLK11 expression was associated with poor overall survival time of CCA patients. The authors also hypothesized the potential signaling pathway of CCDC25 to be associated with KLK11 as involved in metastatic process.

The manuscript is quite interesting, however it was mainly descrptive as the title said and reported data of a small study population (n. 18 patients).

I suggest to increase the study population  (a reliable number of patients is 40). Then it could be interesting to get more insights on the molecular mechanisms linking KLK11 and metastatic process by in vitro experiments and by highlighting literature data.

Author Response

Answer: I am grateful for your suggestion. Under limitation of the time allocated for us to revise the manuscript, we are unable to perform additional experiments using larger samples. We totally agree with your comments and suggestion and that is exactly what we are planning to do as our next step of this series of research.

            For the English improvement, the text has been reviewed and edited by the reviewer of the Publication Clinic of KKU before submission.

Reviewer 2 Report

Cholangiocarcinoma (CCA) represents a diverse group of epithelial cancers united by late diagnosis and poor outcomes. The diagnosis of CCA is usually made through a combination of clinical, biochemical, radiological, and histological information. In addition, coiled-coil domain containing 25 (CCDC25) was reported as a potential therapeutic target for the treatment of patients with CCA.

In this study, the authors found that KLK11 and CCDC25 proteins highly expressed in human CCA tissues, compared to adjacent non-cancerous tissue of a normal bile duct. Furthermore, the expressions of CCDC25 and KLK11 were significantly higher in cancerous tissues than in adjacent tissues of CCA with lymph node but not in CCA without lymph node metastasis. The overall survival curve showed a significantly shorter lifespan in the CCA patients with high expression of KLK11 than in the patients with low expression of KLK11. The author also predicted the interaction between KLK11 and CCDC25 using the bioinformatics tool. This study suggested that KLK11 might serve as a potential prognostic marker for CCA.

Some questions need to be addressed,

  1. The authors showed that “CCDC25 and KLK11 interact with other transcription factors including JAK2, EGFR, mTOR, TGFβ1, VEGF, SMAD2, and MMP9. These proteins were reported as multifunctional proteins in cancers including metastasis.” Did the authors test the expressions of CCDC25 and KLK11-associated transcription factors in human CCA tissues?
  2. The authors showed a correlation between KLK11 and CCDC25. The lifespan was shortened in the CCA patients with high expression of KLK11, compared which in the patients with low expression of KLK11. However, the shorter lifespan was not observed in CCA patients with higher CCDC25 expression than the lower CCDC25 expression. Could the authors please provide some possible reasons?
  3. Some spelling and grammar need to be modified. Line 248, CCDC25 and KLK11 interacts with other transcription factors including JAK2, EGFR, mTOR…

Author Response

1. The authors showed that “CCDC25 and KLK11 interact with other transcription factors including JAK2, EGFR, mTOR, TGFβ1, VEGF, SMAD2, and MMP9. These proteins were reported as multifunctional proteins in cancers including metastasis.” Did the authors test the expressions of CCDC25 and KLK11-associated transcription factors in human CCA tissues?

            Answer: Unfortunately, we have not yet measured those transcriptional factors in human CCA tissues. That is exactly we would like to pursue in our future research.

2. The authors showed a correlation between KLK11 and CCDC25. The lifespan was shortened in the CCA patients with high expression of KLK11, compared which in the patients with low expression of KLK11. However, the shorter lifespan was not observed in CCA patients with higher CCDC25 expression than the lower CCDC25 expression. Could the authors please provide some possible reasons?

            Answer: Thank you for your valuable comment. Although KLK11 was identified as one of the three CCDC25 binding proteins, STRING analysis revealed that interaction of KLK11 and CCDC25 requires some other molecules/pathways. This might account for the discrepancy between KLK11 and CCDC25 expressions and the life span of the patients. This point is discussed at the beginning of the last paragraph of Discussion.

            There are several factors that can influence on the overall survival of CCA patients. Cox proportional hazards model was applied to identify factors that can influence on the overall survival. As shown in Table 1, CCDC25 is not an independent prognostic marker. CCDC25 must consider with age and CEA for prognosis of CCA patients . On the other hand, KLK11 expression was identified as an independent prognostic marker for CCA after multivariate Cox proportional hazard analysis in Table 2.

Table 1. Association of CCDC25, clinicopathological factors and overall survival time by Univariate and Multivariate analysis

Clinicopathological factors

Univariate analysis

Multivariate analysis

HR (95% CI)

P-value

HR (95% CI)

P-value

H-score of CCDC25

(<155 or >155)

1.360(0.475-3.869)

0.567

21.606(1.584-294.630)

0.021**

Sex

(female or male)

1.442 (0.542-3.834)

0.463

0.115(0.007-1.985)

0.137

Age

(<57 or >57 yr.)

1.028 (0.386-2.737)

0.956

0.007(0.000-0.463)

0.020**

ALT

(<36 or >36 U/L)

1.952(0.704-5.416)

0.199

4.731(0.361-62.022)

0.237

AST

(<32 or >32 U/L)

1.520 (0.545-4.239)

0.423

0.110(0.004-2.779)

0.180

ALP

(<121 or >121 U/L)

2.267 (0.746-6.885)

0.149

3.446(0.550-21.588)

0.186

CEA

(<5 or >5 ng/mL)

2.455(0.758-7.955)

0.134

167.841(2.439-11549.011)

0.018**

CA19‑9

(<37 or >37 U/mL)

1.234(0.460-3.307)

0.676

0.726(0.142-3.710)

0.700

Lymph node metastasis

(non-metastasis or metastasis)

1.624(0.460-5.736)

0.451

0.244(0.008-7.697)

0.423

Abbreviations: HR = hazard ratio; CI = confidence interval. **Statistically significant (P<0.05); 95% Confidence Interval.

Table 2. Association of KLK11, clinicopathological factors and overall survival time by Univariate and Multivariate analysis

Clinicopathological factors

Univariate analysis

Multivariate analysis

HR (95% CI)

P-value

HR (95% CI)

P-value

H-score of KLK11

(<185 or >185)

3.076 (0.969-9.765)

0.057*

7.698(1.112-53.310)

0.039**

Sex

(female or male)

1.442 (0.542-3.834)

0.463

0.532(0.078-3.648)

0.521

Age

(<57 or >57 yr.)

1.028 (0.386-2.737)

0.956

0.137(0.008-2.259)

0.165

ALT

(<36 or >36 U/L)

1.952(0.704-5.416)

0.199

2.673(0.193-37.077)

0.464

AST

(<32 or >32 U/L)

1.520 (0.545-4.239)

0.423

0.291(0.015-5.750)

0.417

ALP

(<121 or >121 U/L)

2.267 (0.746-6.885)

0.149

2.468(0.257-23.726)

0.434

CEA

(<5 or >5 ng/mL)

2.455(0.758-7.955)

0.134

10.561(0.671-166.216)

0.094*

CA19‑9

(<37 or >37 U/mL)

1.234(0.460-3.307)

0.676

2.499(0.439-14.217)

0.302

Lymph node metastasis

(non-metastasis or metastasis)

1.624(0.460-5.736)

0.451

0.838(0.053-13.225)

0.900

Abbreviations: HR = hazard ratio; CI = confidence interval. *Statistically significant (P<0.1); 90% Confidence Interval. **Statistically significant (P<0.05); 95% Confidence Interval.

3. Some spelling and grammar need to be modified. Line 248, CCDC25 and KLK11 interacts with other transcription factors including JAK2, EGFR, mTOR…

            Answer: Including the suggested error, the spelling and grammar have been checked by the professional reviewer of the Publication Clinic of KKU. All changes made were marked in red fonts in the text.

Round 2

Reviewer 1 Report

The manuscript has been improved and can be accepted in its present form